# Thermodynamic Analysis of the Landolt-Type Autocatalytic System

**Miloslav Pekař**

Faculty of Chemistry, Brno University of Technology, Purkyňova 118, 612 00 Brno, Czech Republic; pekar@fch.vut.cz

**Abstract:** A recent work demonstrated the example of the Landolt-type reaction system and how the simplest autocatalytic loop is described by the kinetic mass action law and proper parametrization of direct and autocatalytic pathways. Using a methodology of non-equilibrium thermodynamics, the thermodynamic consistency of that kinetic model is analyzed and the mass action description is generalized, including an alternative description by the empirical rate equation. Relationships between independent and dependent reactions and their rates are given. The mathematical modeling shows that following the time evolution of reaction rates provides additional insight into autocatalytic behavior. A brief note on thermodynamic driving forces and coupling with diffusion is added. In summary, this work extends and generalizes the kinetic description of the Landolt-type system, placing it within the framework of non-equilibrium thermodynamics and demonstrating its thermodynamic consistency.

**Keywords:** autocatalysis; Landolt reaction; non-equilibrium thermodynamics; reaction rate; thermodynamic consistency

## 1. Introduction

Autocatalysis refers to the acceleration of a reaction by its product [1–5]. The simplest example is the formal reaction A + X → 2X with the rate (in the forward direction) given by the traditional mass action law as $kc_A c_X$ [3,6]. Autocatalysis is thus essentially a kinetic concept and phenomenon.

Autocatalytic reactions are found in real chemically reacting systems including biological and pharmaceutical applications. Thus, Khot and Pushpavanamcan [7] claim that autocatalysis can be viewed as a mechanism to explain self-replication. They studied a system of two species undergoing autocatalysis to obtain the conditions for extinction or complete conversion of one of the species. The studied system can represent the interconversion of two social groups or isomers into each other. Rich, dynamic behavior including bifurcations and multiplicity of steady states was demonstrated. The results were used to reveal how a mixture of isomers can be driven to a final state consisting of only one isomer. In this way a possible mechanism for obtaining pure enantiomer from a racemic mixture could be suggested.

Woolf [8] included autocatalysis in an article titled "A Hypothesis About the Origin of Biology", which discusses the (potential) mechanisms of that origin. Autocatalysis, besides selection of molecules for linkage by their electrical characteristics or evolution by survival selection, was among the processes that initiated biology. As stated in that work, "autocatalysis is a process where an object or system uses the energy and material resources of the environment to grow itself, or something closely resembling itself." It is claimed that initially autocatalytic processes produced biological membranes. Autocatalysis results in dissipative structures, which result from environmental energy flow and provide order that replaces the order lost in decay processes. To build higher levels of

structure, an autocatalytic system should be coupled to some heterocatalytic action. By linking the autocatalytic system to heterocatalytic action the reaction network of life was able to expand.

Veldhuis et al. [9] describe how an ecosystem organization emerges through ecological autocatalysis. Ecological autocatalysis is analogous to autocatalytic loops in systems chemistry and is claimed to be the backbone of the organization in systems ecology. It forms a set of species populations that promote each other in a loop through positive feedback, and in this way a "core engine" of many ecosystems is created. This is an extension of the original autocatalysis operation in biochemical systems in which reactions catalyze enough substrate for the next reaction, leading to a self-sustaining set of chemical reactions under sufficient input of energy and essential materials [10–12]. As an example of such an autocatalytic loop, the regular and reverse Krebs cycle is given. Autocatalytic processes were theoretically suggested and experimentally verified as a kinetic control mechanism in asymmetric amplification in alkylation reactions by Klussmann et al. [13].

This work, in fact, was motivated by a model analysis published recently by Horváth [14]. He analyzed several typical autocatalytic reaction schemes from the kinetic viewpoint, focusing particularly on the time evolution of the concentration of a product with the autocatalytic action. Among the schemes was a three-step reaction sequence called the Landolt-type autocatalytic system [14]. Horváth stressed the importance of parametrization in autocatalytic routes. This means that, in mass action kinetic models, the ratio of rate constants (coefficients) of the autocatalytic and non-autocatalytic steps is the governing parameter, not the (formal) kinetic order at the autocatalytic specie in the model. Besides kinetics, thermodynamic analysis of reaction systems is also important. The thermodynamic viewpoint was not presented in ref. [14]. Recently, the method of applying a methodology of non-equilibrium thermodynamics to the analysis of autocatalytic processes and their kinetics was shown [15]. This methodology enables the derivation of thermodynamically consistent rate equations which are generalizations of the traditional mass action kinetic law and provides an immediate analysis of thermodynamic consistency of kinetic models. It is based on the permanence of atoms (the atomic structure of the reacting components) and thus belongs to conservative approaches in the terminology used by Érdi and Tóth [16]. It thus does not employ formal chemical species but really stoichiometric equations, though they can be of general form. As noted in ref. [15] this apparently limits the application of that methodology in autocatalysis. How to avoid that limit was illustrated through a formal reaction scheme. In contrast, Horváth discussed schemes representing real reactions.

This work extends both Horváth's and the previous [15] works by thermodynamic analysis of the Landolt-type system studied kinetically in ref. [14]. First, the consistency with non-equilibrium thermodynamics is demonstrated, then the description using independent reactions is discussed; a brief comment on thermodynamic driving force follows. Then, modeling of rate profiles is presented and at the end the relationship of the thermodynamics-based approach to empirical rate equations in autocatalysis is commented on, and a note on diffusion in the Landolt-type system is added.

## 2. Results and Discussion

### 2.1. Thermodynamic Consistency

The Landolt-type autocatalytic scheme was written in ref. [14] purely formally as a set of three steps converting two principal reactants, $A, B$, to one principal product, $C$:

$$A + B \rightarrow C, \tag{1.R1}$$

$$B + C \rightarrow D, \tag{1.R2}$$

$$A + D \rightarrow 2C, \tag{1.R3}$$

To apply the thermodynamic, conservative [16] method, an atom-conserving representation should be found first which can still be general. Perhaps the simplest scheme is as follows:

$$A + B = AB, \tag{2.R1}$$

$$B + AB = AB_2, \tag{2.R2}$$

$$A + AB_2 = 2AB, \tag{2.R3}$$

Clearly, $C \equiv AB, D \equiv AB_2$. The thermodynamic approach used is a mathematical theory, which views stoichiometric equations as equations, therefore, the "=" symbol is used instead of the (double) arrow common in chemistry (the chemical view on kinetics). The autocatalytic step is the third step in both schemes and the autocatalytic specie is $C$ (AB).

The rates written traditionally and restricted to the forward direction only are [14] (cf. scheme (1.R)):

$$r_1 = k_1 c_A c_B; \; r_2 = k_2 c_B c_C; \; r_3 = k_3 c_A c_D. \tag{1}$$

However, the thermodynamic methodology used here begins with finding the rank of the compositional matrix (for details see [17], pp. 150–151; [18,19]). The "atoms" in scheme (2.R) are numbered as 1 = A, 2 = B and components as 1 = A, 2 = B, 3 = AB, 4 = $AB_2$ (the number of components $n = 4$). The compositional matrix thus has the dimension $2 \times 4$ (atoms $\times$ components) and is

$$\|S\| = \begin{bmatrix} 1 & 0 & 1 & 1 \\ 0 & 1 & 1 & 2 \end{bmatrix}. \tag{2}$$

The rank of this matrix is $h = 2$. This means that there are $n - h = 2$ independent reactions in this mixture of four components. The stoichiometric matrix ($\|P\|$) corresponding to independent reactions should fulfill the condition $\|P\|\|S\|^T = \|0\|$ (for details see [18]). It can be easily verified that the following matrix satisfies this condition:

$$\|P\| = \begin{bmatrix} 0 & -1 & -1 & 1 \\ -1 & 0 & 2 & -1 \end{bmatrix}. \tag{3}$$

The two independent reactions corresponding to the matrix (3) are

$$B + AB = AB_2, \tag{3.R1}$$

$$A + AB_2 = 2AB, \tag{3.R2}$$

that is, the last two reactions in scheme (2.R).

The next step is to find rate equations for independent reactions. Non-equilibrium thermodynamics of linear fluids derives the general form of the rate equation as the function $\mathbf{J} = \mathbf{J}(T, \mathbf{c})$ [17] (p. 248); $\mathbf{J}$ is the vector whose components are the rates of independent reactions, $T$ is the temperature and $\mathbf{c}$ is the vector of concentrations. In fact, the well-known kinetic mass action law is thus proved thermodynamically. This general function is approximated by a polynomial of a suitable degree in concentrations with temperature-dependent coefficients. The polynomial is simplified by its application on equilibrium where reaction rate vanishes by definition and where expressions for equilibrium constants can be used. The resulting polynomial, the final form of the rate equation, is called the thermodynamic polynomial [17,19]. In many cases a first or second-degree approximating polynomial is sufficient [17] (p. 249).

The second-degree polynomial leads in the case of two reactions (R3) to the final form of the thermodynamic polynomial (rate equation)

$$\mathbf{J} = \mathbf{k}_{1100}\left(c_A c_B - K_1^{-1} K_2^{-1} c_{AB}\right) + \mathbf{k}_{0110}\left(c_B c_{AB} - K_1^{-1} c_{AB_2}\right) + \mathbf{k}_{1001}\left(c_A c_{AB_2} - K_2^{-1} c_{AB}^2\right). \quad (4)$$

For details see the Supplementary Materials. Here, $\mathbf{J} = (J_1, J_2)$ is the vector of the reaction rates of the two independent reactions, vectors $\mathbf{k}_i$ contain their rate coefficients (constants), for example, $\mathbf{k}_{1100} = (k_{1100}^1, k_{1100}^2)$, $K_p$ refers to their equilibrium constants ($p$ is the index of independent reactions):

$$K_1 = \frac{c_{AB_2,eq}}{c_{B,eq} c_{AB,eq}}, \qquad K_2 = \frac{c_{AB,eq}^2}{c_{A,eq} c_{AB_2,eq}}. \quad (5)$$

Note that from the traditional kinetics viewpoint, the first term in (4) corresponds to the (direct route) step (2.R1) although it is not among the selected independent reactions (3.R). The component rates are related to the (independent) reaction rates as follows:

$$J^A = -J_2, \quad J^B = -J_1, \quad J^{AB} = J_1 + 2J_2, \quad J^{AB_2} = J_1 - J_2. \quad (6)$$

The component rates in the traditional framework given by (1) and used by Horváth are

$$J^A = -r_1 - r_3, \quad J^B = -r_1 - r_2, \quad J^{AB} = r_1 - r_2 + 2r_3, \quad J^{AB_2} = r_2 - r_3 \quad (7)$$

and the consistency of the traditional rate equations with thermodynamic polynomials requires $k_{1001}^1 = k_{0110}^2 = 0$ and

$$k_1 = k_{1100}^1 = k_{1100}^2, \qquad k_2 = k_{0110}^1, \qquad k_3 = k_{1001}^2. \quad (8)$$

Horváth does not consider reversed reaction directions, which means that $K_1^{-1} \to 0$ and $K_2^{-1} \to 0$.

These results are still not a full assessment of the thermodynamic consistency of the Landolt-type system. An additional condition, emanating from the entropic inequality [15], calls for transforming reaction rates to functions of affinities. Reacting mixture corresponding to the Landolt-type scheme has two chemical affinities: $A^1 = -\mu_B - \mu_{AB} - \mu_{AB_2}$, $A^2 = -\mu_A + 2\mu_{AB} - \mu_{AB_2}$ and two constitutional affinities (for details see [20]): $B^1 = (4/3)\mu_A - \mu_B + (1/3)\mu_{AB} - (2/3)\mu_{AB_2}$, $B^2 = -\mu_A + \mu_B + \mu_{AB_2}$ ($\mu_\alpha$ are chemical potentials). The first step of that transformation is writing reaction rates as functions of chemical potentials – details can be found in the Supplementary Materials, here only the final form in affinities is reported:

$$J_1 = k_{1100}^1 \exp\frac{-\mu_A^o - \mu_B^o}{RT} \exp\frac{B^1 + B^2}{RT} \exp\frac{-A^1 - (2/3)A^2}{RT}\left(1 - \exp\frac{A^1 + A^2}{RT}\right)$$
$$+ k_{0110}^1 \exp\frac{-\mu_B^o - \mu_{AB}^o}{RT} \exp\frac{B^1 + (4/3)B^2}{RT} \exp\frac{-(2/3)A^1}{RT}\left(1 - \exp\frac{A^1}{RT}\right), \quad (9.1)$$

$$J_2 = k_{1100}^1 \exp\frac{-\mu_A^o - \mu_B^o}{RT} \exp\frac{B^1 + B^2}{RT} \exp\frac{-A^1 - (2/3)A^2}{RT}\left(1 - \exp\frac{A^1 + A^2}{RT}\right)$$
$$+ k_{1001}^2 \exp\frac{-\mu_A^o - \mu_{AB_2}^o}{RT} \exp\frac{2B^1 + 2B^2}{RT} \exp\frac{-(1/3)A^2}{RT}\left(1 - \exp\frac{A^2}{RT}\right), \quad (9.2)$$

Note that in equilibrium, where $A^p = 0$, the rates are zero as expected. The consistency condition states that that the quadratic form with the symmetric matrix

$$-\begin{bmatrix} \dfrac{\partial J_1}{\partial A^1} & (\dfrac{\partial J_1}{\partial A^2} + \dfrac{\partial J_2}{\partial A^1})/2 \\ & \dfrac{\partial J_2}{\partial A^2} \end{bmatrix}_{eq} \quad (10)$$

is positive semidefinite. This condition finally leads to $k_{1100}^1 \geq 0, k_{0110}^1 \geq 0$ and if we accept that the numbering of independent reactions does not matter, also to $k_{1001}^2 \geq 0$. Referring to (8) it is seen that these results are consistent with the positivity of rate constants in traditional mass-action kinetics.

Horváth states that the kinetic model based on (1) is free of any artificial impurities while able to interpret the autocatalytic feature [14]. Our analysis extends and generalizes his kinetic view by addressing and demonstrating the consistency with non-equilibrium thermodynamics (of linear fluid mixtures [17]).

### 2.2. Independent and Dependent Reactions

The thermodynamic methodology works only with independent reactions which are (mathematically) sufficient to describe mass or molar changes in a reacting mixture. The number of independent reactions in a Landolt-type system, as shown in the preceding section, is equal to two, but the kinetic tradition used also by Horváth operates with rates of three reaction steps $(r_i)$. The relationship between them and the rates of independent reactions $J_p$ is found from the component rates. Comparing (7) and (6), we see that

$$J_1 = r_1 + r_2, \ J_2 = r_1 + r_3, \tag{11}$$

that is, the couple of independent rates cannot be taken from the triple $(r_1, r_2, r_3)$ simply by selecting two of its members. In that triple, the two sums given in (11) represent the two independent rates.

From the practical (experimental) point of view, two independent reactions therefore mean that it is sufficient to measure two rates. Inspecting (6), the simplest way is to measure the component rates of A and B, which are directly related to the two independent rates, $J_1$ and $J_2$.

The (in)dependency of reactions is reflected in the transformation of reaction rates into component rates. This transformation is unequivocal in the case of the two independent rates:

$$\begin{bmatrix} J^A \\ J^B \\ J^{AB} \\ J^{AB_2} \end{bmatrix} = \|P\|^T \begin{bmatrix} J_1 \\ J_2 \end{bmatrix}, \tag{12}$$

whereas in the case of the three (dependent) rates

$$\begin{bmatrix} J^A \\ J^B \\ J^{AB} \\ J^{AB_2} \end{bmatrix} = \|Q\| \begin{bmatrix} r_1 \\ r_2 \\ r_3 \end{bmatrix} \equiv \|Q\| \begin{bmatrix} aJ_1 + bJ_2 \\ (1-a)J_1 - bJ_2 \\ -aJ_1 + (1-b)J_2 \end{bmatrix} \tag{13}$$

with

$$\|Q\| = \begin{bmatrix} -1 & 0 & -1 \\ -1 & -1 & 0 \\ 1 & -1 & 2 \\ 0 & 1 & -1 \end{bmatrix} \tag{14}$$

and arbitrary $a$ and $b$. Combining with (11) we obtain, for example,

$$r_1 = (a/c)r_2 + (b/c)r_3, \qquad c = 1 - a - b. \tag{15}$$

Equation (15) expresses the dependency among $r_i$'s but its coefficients are arbitrary and may be functions of time, because Bowen's approach (see the Methods section) starts with a local balance of mass in which the component rates are such functions.

### 2.3. On Equilibrium and Thermodynamic Driving Forces

Horváth also compares the course of direct transformation of A and B to C (or AB) represented by (1.R1) or (2.R1) with that of autocatalytic pathway represented by (1.R2) and (1.R3) or (2.R2) and (2.R3). A thermodynamic analysis of the direct route was published recently [21]. In this work, the direct route is represented by the term with $k_{1100}^1$ in, for example, (9.1) or (9.2). When the direct route is combined with the autocatalytic route this term contains the product $K_1^{-1}K_2^{-1}$, see (4). This product formally equals the equilibrium constant of (2.R1) and the direct route term in (4) is then formally equal to the rate derived in ref. [21]. However, the equilibrium of the step (2.R1) in scheme (2.R) is controlled by the other two steps.

When reaction rates are expressed as functions of chemical potentials, the term corresponding to the direct route (2.R1) depends on the chemical potentials of A, B, and AB and is identical to the case when only the single transformation A + B = AB occurs [21]. However, in terms of affinities the two cases are different. In the mixture of A, B, AB, and $AB_2$ the term corresponding to (2.R1) contains the two affinities of the two independent reactions (3.R). Thus, the thermodynamic sources ("driving forces") of the kinetics of (apparently) the same reaction are different when this reaction occurs in different reacting mixtures. In other words, in the autocatalytic scheme, the kinetics of the direct route (2.R1) is controlled by the affinities of the autocatalysis-forming steps.

### 2.4. Modeling of Rate Time Profiles

The previous work [14] also discussed (calculated) time profiles of product C concentrations in a batch system for several selected values of rate parameters $k_i$. Because autocatalysis is a kinetic phenomenon, it is very instructive to also inspect rate profiles. Two selected examples are shown in Figures 1 and 2 (and they relate to the black and cyan curves in Figure 1 of [14], respectively). Both figures demonstrate the invalidity of the supposition that reaction (1.R3) is much faster than reactions (1.R1) and (1.R2), which was applied to select the values of $k_i$'s in ref. [14]). This is the consequence of the fact that the reaction rate is affected not only by the value of its rate constant but also of the concentrations.

In Figure 1, the rate of (1.R1) is very low, except in the very beginning, and rates of (1.R2) and (1.R3) are almost identical (the difference not seen in the figure). The autocatalytic effect is seen in the rapid increase of the rate of (1.R3), which evidently also controls the rate of (1.R2) and maintains it at high values. The effect is delayed in some induction period, the length of which can be estimated from the whole course as 5 s, or, at least, about 1 s when the rate of (1.R1) starts to be smaller than the other two rates. Once the supply of reactants is exhausted, all rates drop down to zero forming maxima on $r_2$ and $r_3$ profiles.

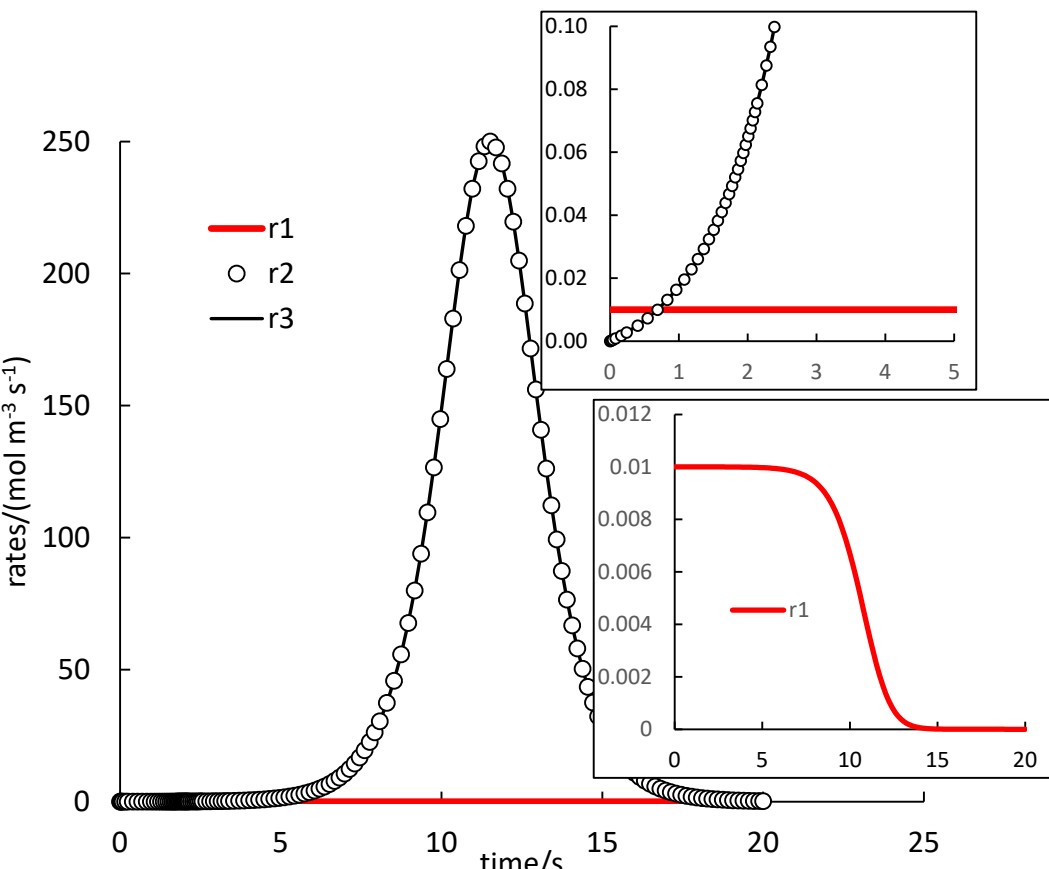

**Figure 1.** Modeled rate time profiles. Reaction scheme (R1), rates (1), batch system. Rate constants correspond to the black curve in Figure 1 in ref. [14]: $k_1 = 10^{-8}$, $k_2 = 10^{-3}$, $k_3 = 10^4$ (all $m^3 \, mol^{-1} \, s^{-1}$). Insets show details in early times (upper) and $r_1$ profile (lower).

In Figure 2, the rate of (1.R1) is the highest for a rather long time. Rates of (1.R2) and (1.R3) are, again, almost identical during the whole course. Here, no induction period is observed and the profiles of the (1.R2), (1.R3) rates seem to always be concave. The auto-catalytic effect can be seen here in increasing rates of (1.R2), (1.R3), while the rate of (1.R1) decreases progressively.

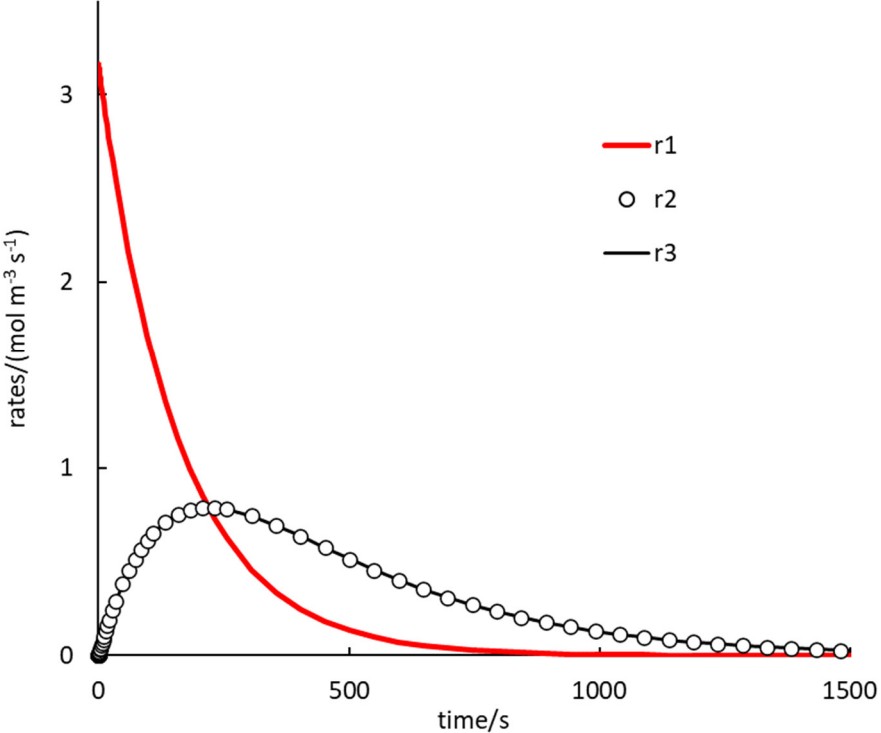

**Figure 2.** Modeled rate time profiles. Reaction scheme (R1), rates (1), batch system. Rate constants correspond to the cyan curve in Figure 1 in [14]: $k_1 = k_2 = 3.1623 \times 10^{-8}$, $k_3 = 10^4$ (all $m^3\,mol^{-1}\,s^{-1}$).

Both figures report on profiles in a batch reactor. In a continuous stirred tank reactor, the situation is rather different and discussed in more detail in the Supplementary Materials. Discussions of autocatalytic kinetics should therefore keep in mind the type of system and consider continuous inflow forcing in flow-through systems.

### 2.5. Empirical Rate Equation

As noted in ref. [14], the rate equation for an autocatalytic reaction is sometimes expressed in an empirical form with an additional term containing the product concentration with positive order. For reaction (1.R1) this means the rate Equation $k_0 c_A c_B + k_a c_A c_B c_C^p$, $p > 0$. The thermodynamic approach used in this work enables a formulation of a very similar equation. Considering only the reaction mixture of A, B, and AB with equilibrium constant $K$, the third degree approximating polynomial leads to the rate equation

$$J = k_{110}(c_A c_B - K^{-1} c_{AB}) + k_{002}(c_{AB}^2 - K c_A c_B c_{AB}).\qquad(16)$$

For a sufficiently high equilibrium constant this equation can be modified to $J = k_{110} c_A c_B - k_{002} K c_A c_B c_{AB}$, which with $k_{002} < 0$ gives the empirical form with $p = 1$.

Alternatively, eq. (16) can be rewritten in the form

$$J = (k_{110} - k_{002} K c_{AB})(c_A c_B - K^{-1} c_{AB}) \equiv k'(c_A c_B - K^{-1} c_{AB}),\qquad(17)$$

which formulates the rate of A + B = AB in terms of AB rate traditional mass action expression but with a concentration (and temperature) dependent rate constant $k'$ (and no presumptions on the magnitude of the equilibrium constant). The autocatalytic effect is hidden in the concentration dependency of $k'$.

*2.6. Note on Diffusion*

When chemical reactions are coupled with diffusion, concentration changes are not caused only by chemical transformations. This problem is outside the scope of this paper, and we add only a brief comparative note. Diffusion can be "self-balanced" which means a special restriction on diffusion fluxes, and it enables using the extent of reaction to describe concentration changes caused by reactions even in coupled reaction-diffusion systems [22].

If the transformation of $A, B$ to $AB$ occurs as a single reaction $A + B = AB$, diffusion is self-balancing when $\sigma_A = \sigma_B = -\sigma_{AB}$ [23]. The symbol $\sigma_\alpha$ is explained in detail in ref. [23] but here it is sufficient to state that it represents the divergence of the diffusion flux of the component $\alpha$.

If the transformation proceeds including intermediate $AB_2$, i. e. according to (R2), diffusion is self-balancing only when the following condition is fulfilled: $\sigma_B = 2\sigma_A + \sigma_{AB}$. Interestingly, there is no direct effect of the diffusion flux of the intermediate $AB_2$ on the self-balancing diffusion condition. On the other hand, the condition was affected by its presence in the $AB$ reacting mixture compared to the case when it is absent. Coupling chemical reactions with diffusion could be common in biological systems. This brief note shows that in a simple chemical transformation, diffusion fluxes can be affected differently due to the presence or absence of an intermediate involved in the autocatalytic scheme.

**3. Methods**

The methodology originated in the previous paper [24] and its general background in non-equilibrium thermodynamics can be found in the book [17]. The core is the general form of the reaction rate as a function of temperature and concentrations, $\mathbf{J} = \mathbf{J}(T, \mathbf{c})$, derived by non-equilibrium thermodynamics for linear fluids which comprise many real systems of interest in (solution and gas) chemistry; $\mathbf{J}$ is the vector of the rates of independent reactions and $\mathbf{c}$ is the vector of concentrations. In other words, the traditional mass-action kinetic law was proved and generalized thermodynamically. The general function is approximated by a polynomial of suitable degree:

$$\mathbf{J} = \sum_{\beta=1}^{Z} \mathbf{k}_{\mathbf{v}_\beta} \prod_{\alpha=1}^{n} c_\alpha^{v_{\beta\alpha}}, \quad \sum_{\alpha=1}^{n} v_{\beta\alpha} \le M. \tag{18}$$

Here, $c_\alpha$ is the molar concentration of component $\alpha$, and $n$ is the total number of components. The vectors $\mathbf{k}_{\mathbf{v}_\beta}$ contain polynomial coefficients dependent on temperature only, the vectors $\mathbf{v}_\beta = (v_{\beta 1}, v_{\beta 2}, \ldots, v_{\beta n})$ contain polynomial powers and are also used as subscripts to index various vectors of polynomial coefficients. For the total number of terms $Z$ see [19,20]. The choice of the polynomial degree is guided by the correspondence between the powers of the polynomial terms and the reaction orders – third-degree, at most, should be appropriate, in many cases first or second-degree is sufficient [17] (p. 249).

The methodology starts with determining the components of a reaction mixture and their atomic composition, which is the basis for finding the number of independent reactions, selecting them and finding the corresponding stoichiometric matrix using the linear algebra approach devised and justified by Bowen [22]. Further steps, briefly, are as follows:

— selection of the degree of the thermodynamic polynomial and writing down the full polynomial,

— some concentrations are expressed from the equilibrium constants of the selected independent reactions and substituted in the equilibrium form of the thermodynamic polynomial,

— restrictions on the polynomial (rate) coefficients follow from the requirement of the general validity of equilibrium [17],

—　these restrictions are introduced into the thermodynamic polynomial, this giving its final, simplified form – the rate equation.

Mathematical modeling was performed with the Chemical Reaction Engineering Module of the COMSOL Multiphysics package, version 5.6.

## 4. Conclusions

A Landolt-type system describing the transformation of two reactants (A, B) to a single product (AB) was recently presented as an example of a kinetic model which is free of any artificial impurities while able to interpret the autocatalytic feature [14]. The analysis of this work extends and generalizes that statement by demonstrating the consistency of the kinetic description of the Landolt-type system with non-equilibrium thermodynamics.

If the transformation of A, B to AB includes an autocatalytic step, its kinetic and diffusional behavior is different from the case when this transformation occurs as a single, direct step. These two routes also differ in their "thermodynamic driving forces" (chemical affinities).

When modeling (autocatalytic) kinetics, it is instructive to follow also the rates of individual reaction steps in the reaction scheme which more immediately reveal the autocatalytic features and the contribution of these steps to the overall transformation reactants to products. Rate time profiles of the same reacting system are different in batch and flow-through reactors due to the forcing of the permanent inflow to the latter system.

**Supplementary Materials:** The following are available online at www.mdpi.com/article/10.3390/catal11111300/s1, file containing details on the derivations described in the main text.

**Funding:** This research received no external funding.

**Data Availability Statement:** No new data were created or analyzed in this theoretical study. Data sharing is not applicable to this article.

**Conflicts of Interest:** The author declares no conflict of interest.

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
