# Peer review of "Thermodynamic Analysis of the Landolt-Type Autocatalytic System"

_catalysts, doi:10.3390/catal11111300_

Round 1

Reviewer 1 Report

The kinetics working in the Landolt-type autocatalytic system is described. The author's motivation may be the development of the theoretical argument reported by Horváth (2020), by applying the matrix-type analysis method, which is probably established by Bowen (1968). As is evident from the fact that Bowen's paper is rarely cited except by Bowen and the authors of this manuscript, the methods in the manuscript are too technical to be understood by ordinary readers. This difficulty is the same for the reviewer, so there is little space for the reviewer to contribute to the publication of this manuscript.

The introduction is described well. The significance of the autocatalysis in our world is clearly explained. On the other hand, from the introduction of the “compositional matrix” (eq.2) and “stoichiometric matrix” (eq.3), the description becomes specialized. The reviewers assume that the author can explain more kindly. For example, why the author used this method and a description of T (line 98), and the addition of a method to multiply the matrix (S and P) by the equation in line 98.

In addition, the conclusion and novelty of the study should be clarified. The statement ll. 74-76 and the conclusions-section, there is no information on what the authors have developed in this study. The author claims that this work is an extension of both Horváth and the authors previous study, but readers don't understand what extensions are there. The reviewer is afraid that the author only complicated Horváth’s claims using the matrix method.

Overall, the paper may be published from Catalyst, but there is a large room to improve the paper by clarifying the motivation, methods, and the conclusions of the study.

Reviewer 2 Report

Attached pdf
